# Localized TabICLv2: Scaling Tabular In-Context Learning through k-NN

**Beimnet Bekele Guta** [1]

## Abstract

Foundational models for tabular data have made significant progress in recent years, with TabICLv2 reporting state-of-the-art performance on several tabular classification tasks. However, full-context tabular ICL still suffers from attention cost that grows with the training-context size, which limits its ability to handle large datasets efficiently. Localized TabICLv2 introduces a method that reduces the inference cost of TabICLv2 by retrieving only the $k$ nearest training neighbours for each test point, measured by similarity in the model's Stage 2 row-representation space, rather than using the full training context. This requires no architectural changes, and we show that accuracy retention can be improved through additional Stage 2 and Stage 3 fine-tuning. On TabArena classification tasks, the fine-tuned localized model retains 98.64% of Full TabICLv2 accuracy and it achieves a median 2.18× speedup in batch inference, and reaches approximately 249× median speedup in the single-query serving setting.

## 1. Introduction

Tabular machine learning has been dominated by gradient-boosted decision tree methods such as XGBoost (Chen & Guestrin, 2016) and LightGBM (Ke et al., 2017), which are highly effective (Grinsztajn et al., 2022) but require dataset-specific training and tuning (Koster & Sigrist, 2026). Recent tabular transformer models such as TabPFN (Hollmann et al., 2023), TabICL (Qu et al., 2025), and TabICLv2 (Qu et al., 2026) address this limitation by using in-context learning to make predictions from training and query rows in a single forward pass.

Despite these advances, TabICLv2 still exhibits inherent scalability limitations. In particular, during Stage 3 (TFicl),

---

[1]University of Cambridge, Cambridge, United Kingdom. Correspondence to: Beimnet Bekele Guta <bbg25@cam.ac.uk, beimnetbekele1234@gmail.com>.

*Proceedings of the $2^{nd}$ ICML Workshop on Foundation Models for Structured Data*, Seoul, South Korea. 2026. Copyright 2026 by the author(s).

each test datapoint attends to all training rows via full self-attention. As the size of the training dataset increases, both computational cost and memory requirements grow rapidly. This raises a natural question: can we restrict the context to a subset of training examples that are comparatively more informative for the prediction task? These observations motivate our approach, which focuses on limiting the inference-time context to the most relevant training instances.

We propose Localized TabICLv2, which retrieves the top-$k$ most similar training rows for each test datapoint using the learned Stage 2 representation space, and then performs standard TabICL inference using only this local context. We further fine-tune Stage 2 and Stage 3 to improve representation quality and maintain accuracy under localized inference. Empirically, our method retains up to 98.64% of the accuracy of Full TabICLv2 on TabArena (Erickson et al., 2025), while achieving a median 2.18× speedup in batch inference and a median 249× speedup in single-query serving.

Our contributions are as follows: **(I)** We show that TabICLv2 can be converted into an efficient localized inference model by retrieving neighbours in its learned Stage 2 representation space. **(II)** We demonstrate the impact of jointly fine-tuning Stage 2+3, leading to improved retrieval quality and better accuracy retention. **(III)** We provide extensive evaluation showing this approach achieves high accuracy retention while substantially reducing inference cost.

## 2. Related Work

Recent tabular foundation models use in-context learning (ICL) to perform tabular prediction in a single forward pass without task-specific parameter updates. TabPFN was one of the first transformer-based models in this direction, using synthetic pretraining to approximate Bayesian inference, but it remains limited to smaller datasets (Hollmann et al., 2023).

TabICL addresses this limitation by constructing fixed-dimensional row embeddings before performing ICL, allowing transformers to jointly process training and test rows more effectively (Qu et al., 2025). TabICLv2 builds on this by further improving performance through optimised pretraining and architectural innovations such as scalable soft-

max attention (Qu et al., 2026). This model achieves state-of-the-art results on multiple tabular benchmarks. However, full-context inference remains costly in the final transformer stage, where each query attends to all training examples.

Retrieval-based strategies have been explored as a way to alleviate these constraints. Methods such as LoCalPFN use k-nearest neighbour (k-NN) retrieval to construct a localized context for each query point (Thomas et al., 2024). This has been shown to improve model expressivity. However, LoCalPFN relies on a fixed pretrained representation space and is designed around the smaller-data regime of TabPFN. As a result, the retrieval step is not explicitly optimised for the downstream ICL task or for the large-context setting in which TabICLv2 operates. While promising, the systematic combination of retrieval with targeted fine-tuning remains less explored in the tabular domain compared to the success of retrieval-augmented methods in natural language processing (Lewis et al., 2020).

In other domains, efficient and localized attention mechanisms such as Longformer, BigBird, and Swin Transformers reduce complexity by focusing on local or hierarchical dependencies (Beltagy et al., 2020; Zaheer et al., 2020; Liu et al., 2021).

Drawing inspiration from these developments, we adapt the idea of locality to TabICLv2. Specifically, our approach combines k-NN-based localization with a jointly fine-tuned representation space and ICL component, enabling the model to learn embeddings that are better suited for retrieval while improving accuracy retention under localized inference. The limitations of full-context models motivate this approach, which aims to improve scalability while preserving strong predictive performance on tabular tasks.

## 3. Methodology

**Overview**: Localized TabICLv2 follows a similar inference strategy to TabICL, with three stages. The first two stages produce row representations through distribution-aware column-wise embedding and row-wise interaction modeling. Stage 3 then performs in-context prediction by using labelled support rows to predict labels for query rows (Qu et al., 2025; 2026).

We introduce a k-NN retrieval method between Stage 2 and Stage 3. At Stage 2, each row is represented in a learned vector space. We leverage these embeddings to implement cosine-similarity-based k-NN retrieval, selecting the training rows closest to each test datapoint. This limits the context passed to the ICL predictor, while aiming to retain predictive performance close to that of the full-context model.

**Detailed Implementation**: We begin by initializing the TabICLv2 classifier with `kv_cache = 'repr'` and running the `fit()` method on the training data. Under this setting, Stage 1 and Stage 2 are executed once, and the resulting row representations are cached in `model_kv_cache_`. This allows the model to reuse precomputed embeddings without recomputing them during inference.

At prediction time, embeddings for the test datapoints are generated using the cached column representations. These embeddings are then used to retrieve relevant training examples.

Using these representations, we perform k-nearest neighbour (k-NN) retrieval to select the top-$k$ most similar training datapoints for each test instance. Similarity is computed in the learned representation space through cosine similarity. This constructs a localized context tailored to each query.

Finally, inference is performed using Stage 3, following the standard TabICL procedure. The key difference is that, instead of using the full training set, only the retrieved $k$ training datapoints are provided as context. This significantly reduces the effective sequence length while preserving prediction behaviour.

Both Full TabICLv2 and Localized TabICLv2 share the same cached Stage 1–2 representations, so the relative speedup mainly comes from reducing the Stage 3 context. This changes the dominant cost from full-context attention to fixed local-context attention. In practice, this theoretical reduction does not translate directly into wall-clock speedup because retrieval, batching, sequence length, memory movement, and kernel overhead also affect runtime. Full derivations and systems-level discussion are provided in Appendix A.

**Stage 2–3 Fine-tuning**: While the inference scheme works well for improving inference speed, the model was not originally trained in such a setting, which can lead to a drop in accuracy. To compensate for this, we further fine-tune both Stage 2 and Stage 3 of TabICLv2.

We start from the pretrained TabICLv2 weights. Since Stage 1 is responsible for learning column embeddings, it is kept frozen. In contrast, we fine-tune Stage 2 (row representations) and Stage 3 (ICL predictor) to improve representation quality for k-NN retrieval.

We train using synthetic data generated from the `mix_scm` prior, with sequence lengths between 10k and 30k sampled log-uniformly. The batch size is 64, with one gradient step per synthetic table. We use the AdamW optimiser (Loshchilov & Hutter, 2019) with differential learning rates: a lower learning rate for Stage 2 ($2 \times 10^{-7}$) to preserve pretrained representations, and a higher learning rate for Stage 3 ($2 \times 10^{-6}$). Gradients are clipped with a maximum

norm of 1.0, and a constant learning rate schedule is used.

# 4. Results and Discussion

We evaluate Localized TabICLv2 on the TabArena benchmark (Erickson et al., 2025), covering 38 binary and multiclass classification datasets. We use an 80/20 stratified train/test split and run each experiment over three random seeds. Binary tasks are evaluated using ROC AUC, multiclass tasks using log-loss, and accuracy is reported as an overall summary metric. Unless stated otherwise, all localized methods use $k = 32$ nearest neighbours. All experiments are run on an NVIDIA A100 GPU.

We compare Full TabICLv2, pretrained Localized TabI-CLv2, S2+S3 fine-tuned Localized TabICLv2, XGBoost trained on the full training set and retrieval-only baselines, including XGBoost trained on the retrieved k-nearest neighbours and k-NN majority vote. Full TabICLv2 serves as the full-context reference model, while the pretrained localized model isolates the effect of Stage 2+3 fine-tuning. Speedup is reported relative to Full TabICLv2, and accuracy retention is measured as the ratio between Localized TabICLv2 accuracy and Full TabICLv2 accuracy.

**K-Sensitivity**: We start by measuring the impact of retrieval size on the accuracy–speed trade-off of localization. We use pretrained TabICLv2 weights without any fine-tuning and study how varying $k$ affects both accuracy retention and speedup. The evaluation is conducted on 8 datasets from TabArena, spanning a dataset size range of (13K–285K), using an 80/20 train–test split.

*Table 1.* Aggregate results vs Full TabICLv2.

| $k$ | Acc retention (%) | Mean pred (s) | Mean speedup | Median speedup |
|---|---|---|---|---|
| 16 | 96.93 | 6.72 | 2.77× | 3.04× |
| 32 | 97.57 | 9.19 | 1.97× | 2.14× |
| 64 | 98.40 | 15.61 | 1.14× | 1.11× |
| 128 | 98.90 | 27.22 | 0.66× | 0.66× |

Clear trends can be observed. As $k$ increases, accuracy retention improves monotonically, while speedup gains diminish. This behavior is expected: a larger context window provides more evidence for the model to infer accurate predictions, but it also increases attention computation cost, reducing efficiency.

One important detail is that, since the test split is 20% of the dataset, localization repeatedly selects $k$ training rows per query, leading to duplication of training examples across queries. This introduces additional GPU overhead and limits the achievable speedup.

Overall, $k = 32$ provides a practical balance, achieving

97.57% accuracy retention with 1.97× mean speedup. For this reason, we adopt $k = 32$ as the default setting in subsequent fine-tuning experiments.

**TabArena Accuracy Evaluation**: After the k-sensitivity sweep, we evaluate all main methods on the full TabArena benchmark. This experiment measures whether Localized TabICLv2 can retain the predictive performance of Full TabICLv2 and how fine-tuning affects accuracy retention. We also compare two retrieval spaces: **Embed**, which retrieves neighbours using TabICLv2's learned Stage 2 row representations, and **Raw**, which retrieves neighbours directly in the original feature space. The mean ROC AUC, and Log-loss (where applicable) are reported in Table 2. Detailed per-dataset TabArena evaluation results are provided in Appendix B.

*Table 2.* Main TabArena evaluation across 38 datasets. Results are reported as macro-mean with standard deviation in parentheses. AUC is ROC AUC over 30 binary datasets; Log-Loss is over 8 multiclass datasets. Embed and Raw use Stage 2 embedding-space and raw feature-space retrieval, respectively. PT denotes pretrained, FT denotes S2+S3 fine-tuned, and $k = 32$.

| Method | AUC | Log-Loss | Accuracy |
|---|---|---|---|
| Full TabICLv2 | 0.855 (0.090) | 0.253 (0.198) | 0.885 (0.086) |
| Embed PT | 0.809 (0.107) | 0.339 (0.252) | 0.861 (0.097) |
| Embed FT | 0.827 (0.096) | 0.293 (0.223) | 0.873 (0.090) |
| Raw PT | 0.798 (0.114) | 0.365 (0.264) | 0.857 (0.098) |
| Raw FT | 0.817 (0.105) | 0.319 (0.210) | 0.872 (0.090) |
| XGBoost | 0.826 (0.105) | 0.340 (0.261) | 0.873 (0.093) |

Compared with Raw FT, Embed FT achieves better AUC, log-loss, and overall accuracy, suggesting that TabICLv2's learned Stage 2 representation space provides a better retrieval basis than raw features.

Overall, the localized method does not surpass the accuracy of Full TabICLv2. However, the performance drop is relatively small and remains promising given the efficiency gains. S2+S3 fine-tuning reduces the gap compared to the pretrained localized model by +1.18 percentage points (accuracy: $0.8613 \rightarrow 0.8731$), corresponding to recovering approximately 49.5% of the gap to Full TabICLv2.

For binary classification, the ROC AUC of the S2+S3 fine-tuned model exceeds XGBoost (0.8271), showing that even with a small context window, the localized model can reach tree-based performance without hyperparameter tuning. For multiclass tasks, log-loss follows a similar trend: the fine-tuned localized model outperforms XGBoost while remaining slightly below Full TabICLv2.

In terms of overall retention, pretrained localized inference achieves 97.31% of Full TabICLv2 accuracy, which

increases to 98.64% after fine-tuning. The fine-tuned model outperforms the pretrained localized variant on 33 out of 38 datasets, indicating consistent gains.

**Batch-Inference Speed Evaluation**: We first evaluate batch inference speed on 15 large TabArena datasets. Localized TabICLv2 achieves a median speedup of $2.18\times$ under an 80/20 train–test split. However, the speedup is not strictly monotonic across datasets, due to differences in task structure, feature dimensionality, and GPU memory behaviour. Full per-dataset results are reported in Appendix D.

To better isolate the relationship between dataset size and speedup, we analyse the credit-card-fraud dataset from OpenML (OpenML, 2018) under varying training sizes while keeping other factors constant. Here, a clear monotonic trend emerges, with speedup increasing as $N_{train}$ grows, confirming the expected scaling trend: localization becomes more beneficial for larger training sets.

*Table 3.* Credit-card-fraud dataset under varying training sizes.

| $N_{train}$ | $N_{test}$ | Total | Full (s) | Loc (s) | Speedup |
|---|---|---|---|---|---|
| 10,000 | 2,500 | 12,500 | 0.91 | 0.85 | $1.07\times$ |
| 25,000 | 6,250 | 31,250 | 2.76 | 2.18 | $1.27\times$ |
| 50,000 | 12,500 | 62,500 | 7.39 | 4.55 | $1.63\times$ |
| 100,000 | 25,000 | 125,000 | 22.97 | 9.96 | $2.31\times$ |
| 150,000 | 37,500 | 187,500 | 45.96 | 16.59 | $2.77\times$ |
| 200,000 | 50,000 | 250,000 | 75.93 | 23.66 | $3.21\times$ |
| 227,845 | 56,962 | 284,807 | 96.60 | 27.89 | $3.46\times$ |

**Limited-Query Latency**: We evaluate the method in a deployment setting by limiting the number of test queries to small batches, simulating real-time usage. This allows us to measure per-query latency and assess whether localization makes TabICL practical for low-latency serving.

We select five values for the number of test queries (1, 5, 10, 50, 100) and evaluate the per-query latency across six datasets with a range of training dataset sizes.

*Table 4.* Limited-query latency.

| $N_{test}$ | Full TabICLv2 Latency (ms/query) | Localized TabICLv2 Latency (ms/query) | Approx. Speedup |
|---|---|---|---|
| 1 | $\sim$76,000 | $\sim$59 | $\sim$249$\times$ |
| 5 | $\sim$15,000 | $\sim$12 | $\sim$250$\times$ |
| 10 | $\sim$7,500 | $\sim$6 | $\sim$248$\times$ |
| 50 | $\sim$1,500 | $\sim$1.3 | $\sim$229$\times$ |
| 100 | $\sim$750 | $\sim$0.8 | $\sim$192$\times$ |

For a single query, a maximum speedup of $1223\times$ is achieved, with a median speedup of approximately $249\times$ across the six datasets. This highlights the effectiveness of

localization in low-query settings. Additional per-dataset speedups and runtime measurements are provided in Appendix E.

This behaviour arises because Full TabICLv2 must pass all $N_{train}$ tokens through attention for every prediction call, making its latency dependent on the training dataset size. In contrast, Localized TabICLv2 limits the context to a fixed $k$-sized subset after retrieval, leading to lower and more stable per-query latency ($\sim$59 ms compared to $\sim$76 s at $N \approx 228K$). We further analyse this effect using the credit-card-fraud dataset under varying dataset sizes, with results shown in Appendix F.

**ICL Head Ablation (38 TabArena Datasets)**: To better understand whether the observed accuracy retention is driven by the combined effect of the ICL head and the KNN retrieval step, we conduct an ablation test. We compare the localized TabICLv2 against retrieval-only baselines using the same $k = 32$ neighbours, including XGBoost trained on the retrieved subset and a k-NN majority-vote classifier. Per-dataset evaluation results are provided in Appendix G.

*Table 5.* Win rates: Localized TabICLv2 vs retrieval-only baselines.

| Baseline | Loc TabICLv2 win rate |
|---|---|
| XGBoost trained on k-NN | 37 / 38 |
| k-NN majority vote | 37 / 38 |

Localized TabICLv2 wins against both XGBoost-on-k-NN and k-NN majority vote on 37/38 of the datasets. Since these baselines use identical retrieved contexts but lack an ICL head, this result indicates that the ICL component contributes beyond simple local label aggregation.

## 5. Conclusion

Localized TabICLv2 limits the context given to the ICL head by retrieving only the most relevant training examples through k-NN on a learned representation space. This reduces the per-query Stage 3 attention context from the full training set to only $k$ retrieved neighbours, while preserving most of the predictive performance on TabArena through Stage 2+3 fine-tuning.

Overall, the results show that tabular foundation models can be made more efficient, making them more practical for real-world deployment. However, localization may underperform when useful evidence is not captured by the retrieved neighbours, and retrieval overhead can reduce speedups on smaller datasets or large query batches.

Future work can extend the evaluation to regression tasks, compare directly with LoCalPFN, and study FAISS-style retrieval backends, alongside adaptive $k$, learned retrieval metrics, and improved fine-tuning strategies.

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

# A. Theoretical Analysis of Speedup

Both Full TabICLv2 and Localized methods use the same KV-cache mechanism: training representations are computed once during `fit()`, and at prediction time Stages 1–2 are applied only to test data. Therefore, the computational cost of Stages 1–2 is identical for both methods and does not contribute to the relative speedup.

The overall inference cost can be decomposed as:

$$T_{\text{full}} = T_{S1+S2} + T_{S3}^{\text{full}}$$
$$T_{\text{local}} = T_{S1+S2} + T_{\text{retrieval}} + T_{S3}^{\text{local}}$$

which gives the speedup:

$$\text{Speedup} = \frac{T_{S1+S2} + T_{S3}^{\text{full}}}{T_{S1+S2} + T_{\text{retrieval}} + T_{S3}^{\text{local}}}$$

Since $T_{S1+S2}$ is shared, the difference arises from Stage 3 which forms the main bottleneck in TabICLv2 (Qu et al., 2026) and the retrieval cost.

## A.1. Complexity Analysis

Let $N_{\text{train}}$ denote the number of training rows, $N_{\text{test}}$ denote the number of test rows, and $k$ denote the number of retrieved neighbours.

In Full TabICLv2, Stage 3 processes a sequence of length:

$$T_{S3}^{\text{full}} = O(N_{\text{train}}^2 + N_{\text{train}} N_{\text{test}})$$

where $N_{\text{train}}^2$ represents attention within the training dataset and $N_{\text{train}} N_{\text{test}}$ attention between each test datapoint to the training dataset

In Localized TabICLv2, each test point is processed independently with a fixed context of size k, resulting in:

$$T_{S3}^{\text{local}} = O(N_{\text{test}} k^2)$$

Considering only the Stage 3 ICL component, this suggests a theoretical reduction factor of:

$$\frac{T_{S3}^{\text{full}}}{T_{S3}^{\text{local}}} \approx \frac{N_{\text{train}}(N_{\text{train}} + N_{\text{test}})}{N_{\text{test}} k^2}$$

This shows that when $N_{\text{test}} \ll N_{\text{train}}$ (a common setting during inference), the localized method can achieve substantial speed improvements. The gains scale with both the size of the training set and the number of test datapoints.

However, this theoretical reduction does not directly translate to wall-clock speedup due to retrieval cost, hardware and implementation effects.

First, the two methods differ in how computation is structured: Full TabICLv2 executes a single forward pass over a long sequence, whereas Localized TabICLv2 executes $N_{\text{test}}$ forward passes over short sequences of length $k + 1$. Modern attention kernels (e.g. FlashAttention (Dao et al., 2022)) are optimised for large sequence lengths, where computation can be efficiently parallelised. For smaller sequences, kernel launch overhead and poor occupancy reduce effective throughput limiting the speed gain. Moreover, Full TabICLv2 benefits from efficient batching when the number of test datapoints is large while Localized TabICLv2 incurs repeated per-query overhead, including indexing, stacking, and data movement costs.

$$T_{S3}^{\text{local}} \approx N_{\text{test}} \cdot \left( O(k^2) + \epsilon_{\text{overhead}} \right)$$

where $\epsilon_{\text{overhead}}$ includes indexing, stacking, and data movement costs. As a result, Localized TabICLv2 achieves the greatest speedup when the training set is large or when the number of test queries is small.

# B. TabArena Evaluation

*Table 6.* Per-dataset TabArena primary-metric results comparing Full TabICLv2, Localized TabICLv2 with Stage 2+3 fine-tuning, pretrained Localized TabICLv2, raw pretrained Localized TabICLv2, raw Stage 2+3 fine-tuned Localized TabICLv2, and XGBoost. Binary tasks report ROC AUC, where higher is better; multiclass tasks report log-loss, where lower is better.

| Dataset | Task | Classes | Metric | Full | S2+S3 | Pretrain | Raw PT | Raw FT | XGBoost |
|---|---|---|---|---|---|---|---|---|---|
| APSFailure | binary | – | ROC AUC | 0.9949 | 0.9911 | 0.9911 | 0.9930 | 0.9929 | 0.9895 |
| Amazon_employee_access | binary | – | ROC AUC | 0.8587 | 0.8247 | 0.8231 | 0.7591 | 0.7636 | 0.8301 |
| Bank_Customer_Churn | binary | – | ROC AUC | 0.8709 | 0.8409 | 0.8115 | 0.8169 | 0.8334 | 0.8426 |
| Bioresponse | binary | – | ROC AUC | 0.8605 | 0.8082 | 0.8268 | 0.8492 | 0.8364 | 0.8702 |
| Diabetes130US | binary | – | ROC AUC | 0.6667 | 0.6390 | 0.6163 | 0.5796 | 0.5934 | 0.6412 |
| E-CommerceShipping | binary | – | ROC AUC | 0.7470 | 0.7392 | 0.7294 | 0.7280 | 0.7366 | 0.7289 |
| Fitness_Club | binary | – | ROC AUC | 0.8150 | 0.8079 | 0.7948 | 0.7900 | 0.7947 | 0.7570 |
| GiveMeSomeCredit | binary | – | ROC AUC | 0.8685 | 0.8460 | 0.8411 | 0.8240 | 0.8301 | 0.8562 |
| HR_Analytics | binary | – | ROC AUC | 0.7994 | 0.7864 | 0.7565 | 0.7437 | 0.7786 | 0.7816 |
| Is_good_customer | binary | – | ROC AUC | 0.7541 | 0.7305 | 0.6608 | 0.6390 | 0.6928 | 0.6807 |
| MIC | multiclass | 8 | Log-loss | 0.5116 | 0.5943 | 0.7460 | 0.7977 | 0.5892 | 0.6982 |
| Marketing_Campaign | binary | – | ROC AUC | 0.9301 | 0.8881 | 0.8600 | 0.8487 | 0.8764 | 0.9084 |
| NATICUSdroid | binary | – | ROC AUC | 0.9890 | 0.9860 | 0.9820 | 0.9779 | 0.9832 | 0.9865 |
| SDSS17 | multiclass | 3 | Log-loss | 0.0690 | 0.1043 | 0.1199 | 0.1370 | 0.1222 | 0.0765 |
| airline_satisfaction | binary | – | ROC AUC | 0.9953 | 0.9874 | 0.9898 | 0.9878 | 0.9878 | 0.9935 |
| anneal | multiclass | 5 | Log-loss | 0.0107 | 0.0243 | 0.0212 | 0.0292 | 0.0333 | 0.0223 |
| bank-marketing | binary | – | ROC AUC | 0.7741 | 0.7412 | 0.7217 | 0.7229 | 0.7440 | 0.7580 |
| blood-transfusion | binary | – | ROC AUC | 0.7727 | 0.7244 | 0.7091 | 0.7126 | 0.7403 | 0.6754 |
| churn | binary | – | ROC AUC | 0.9313 | 0.9150 | 0.9128 | 0.9121 | 0.9155 | 0.9116 |
| coil2000 | binary | – | ROC AUC | 0.7788 | 0.7034 | 0.6843 | 0.6742 | 0.6981 | 0.7084 |
| credit-g | binary | – | ROC AUC | 0.7654 | 0.7437 | 0.6687 | 0.6875 | 0.7441 | 0.7338 |
| credit_card_default | binary | – | ROC AUC | 0.7911 | 0.7737 | 0.7388 | 0.7280 | 0.7669 | 0.7634 |
| diabetes | binary | – | ROC AUC | 0.8558 | 0.8320 | 0.8027 | 0.7912 | 0.8273 | 0.8273 |
| hazelnut_contaminant | binary | – | ROC AUC | 0.9955 | 0.9731 | 0.9735 | 0.9707 | 0.9729 | 0.9795 |
| heloc | binary | – | ROC AUC | 0.8014 | 0.7670 | 0.7197 | 0.7257 | 0.7805 | 0.7792 |
| hiva_agnostic | multiclass | 3 | Log-loss | 0.2070 | 0.2097 | 0.2421 | 0.2406 | 0.2070 | 0.2904 |
| in_vehicle_coupon | binary | – | ROC AUC | 0.8607 | 0.8282 | 0.7892 | 0.7529 | 0.7914 | 0.8371 |
| jm1 | binary | – | ROC AUC | 0.7811 | 0.7627 | 0.7566 | 0.7567 | 0.7658 | 0.7224 |
| kddcup09_appetency | binary | – | ROC AUC | 0.8139 | 0.7409 | 0.7372 | 0.6341 | 0.6415 | 0.7579 |
| maternal_health_risk | multiclass | 3 | Log-loss | 0.3407 | 0.3949 | 0.4015 | 0.4046 | 0.3987 | 0.4525 |
| online_shoppers | binary | – | ROC AUC | 0.9425 | 0.9283 | 0.9158 | 0.9017 | 0.9214 | 0.9236 |
| polish_bankruptcy | binary | – | ROC AUC | 0.9534 | 0.8800 | 0.8720 | 0.8520 | 0.8619 | 0.9499 |
| qsar-biodeg | binary | – | ROC AUC | 0.9447 | 0.9345 | 0.9291 | 0.9308 | 0.9344 | 0.9270 |
| seismic-bumps | binary | – | ROC AUC | 0.7794 | 0.7481 | 0.7275 | 0.7095 | 0.7428 | 0.7237 |
| splice | multiclass | 3 | Log-loss | 0.0956 | 0.1241 | 0.2139 | 0.2592 | 0.2514 | 0.1201 |
| student_dropout | multiclass | 3 | Log-loss | 0.5372 | 0.6119 | 0.6564 | 0.6912 | 0.6222 | 0.6738 |
| taiwanese_bankruptcy | binary | – | ROC AUC | 0.9579 | 0.9409 | 0.9298 | 0.9274 | 0.9453 | 0.9382 |
| website_phishing | multiclass | 3 | Log-loss | 0.2548 | 0.2783 | 0.3110 | 0.3590 | 0.3282 | 0.3858 |

*Table 7.* Per-dataset TabArena accuracy results comparing Full TabICLv2, Localized TabICLv2 with Stage 2+3 fine-tuning, pretrained Localized TabICLv2, raw pretrained Localized TabICLv2, raw Stage 2+3 fine-tuned Localized TabICLv2, and XGBoost.

| Dataset | Task | Full | S2+S3 | Pretrain | Raw PT | Raw FT | XGBoost |
|---|---|---|---|---|---|---|---|
| APSFailure | binary | 0.9955 | 0.9932 | 0.9938 | 0.9938 | 0.9932 | 0.9942 |
| Amazon_employee_access | binary | 0.9519 | 0.9483 | 0.9482 | 0.9431 | 0.9473 | 0.9491 |
| Bank_Customer_Churn | binary | 0.8635 | 0.8532 | 0.8238 | 0.8407 | 0.8525 | 0.8518 |
| Bioresponse | binary | 0.7883 | 0.7390 | 0.7554 | 0.7772 | 0.7608 | 0.8034 |
| Diabetes130US | binary | 0.9121 | 0.9120 | 0.8966 | 0.8636 | 0.9120 | 0.9111 |
| E-CommerceShipping | binary | 0.6838 | 0.6644 | 0.6495 | 0.6474 | 0.6655 | 0.6453 |
| Fitness_Club | binary | 0.7589 | 0.7600 | 0.7700 | 0.7411 | 0.7611 | 0.7322 |
| GiveMeSomeCredit | binary | 0.9373 | 0.9361 | 0.9332 | 0.9320 | 0.9353 | 0.9350 |
| HR_Analytics | binary | 0.7928 | 0.7844 | 0.7615 | 0.7554 | 0.7734 | 0.7818 |
| Is_good_customer | binary | 0.8899 | 0.8879 | 0.8454 | 0.8348 | 0.8860 | 0.8725 |
| MIC | multiclass | 0.8627 | 0.8627 | 0.8461 | 0.8333 | 0.8578 | 0.8647 |
| Marketing_Campaign | binary | 0.8966 | 0.8832 | 0.8624 | 0.8668 | 0.8780 | 0.8869 |
| NATICUSdroid | binary | 0.9509 | 0.9433 | 0.9388 | 0.9348 | 0.9435 | 0.9460 |
| SDSS17 | multiclass | 0.9786 | 0.9739 | 0.9732 | 0.9689 | 0.9699 | 0.9759 |
| airline_satisfaction | binary | 0.9628 | 0.9392 | 0.9490 | 0.9473 | 0.9426 | 0.9562 |
| anneal | multiclass | 0.9981 | 0.9981 | 0.9981 | 0.9981 | 0.9981 | 0.9944 |
| bank-marketing | binary | 0.8945 | 0.8919 | 0.8835 | 0.8825 | 0.8913 | 0.8919 |
| blood-transfusion | binary | 0.8089 | 0.7778 | 0.7533 | 0.7622 | 0.7933 | 0.7289 |
| churn | binary | 0.9700 | 0.9463 | 0.9367 | 0.9357 | 0.9427 | 0.9550 |
| coil2000 | binary | 0.9399 | 0.9393 | 0.9250 | 0.9220 | 0.9386 | 0.9325 |
| credit-g | binary | 0.7633 | 0.7333 | 0.6783 | 0.6867 | 0.7417 | 0.7283 |
| credit_card_default | binary | 0.8226 | 0.8201 | 0.7992 | 0.7952 | 0.8159 | 0.8142 |
| diabetes | binary | 0.7619 | 0.7597 | 0.7511 | 0.7424 | 0.7554 | 0.7684 |
| hazelnut_contaminant | binary | 0.9653 | 0.9118 | 0.9194 | 0.9153 | 0.9118 | 0.9285 |
| heloc | binary | 0.7337 | 0.7194 | 0.6708 | 0.6643 | 0.7156 | 0.7173 |
| hiva_agnostic | multiclass | 0.9649 | 0.9649 | 0.9649 | 0.9649 | 0.9649 | 0.9623 |
| in_vehicle_coupon | binary | 0.7844 | 0.7584 | 0.7351 | 0.7010 | 0.7286 | 0.7600 |
| jm1 | binary | 0.8184 | 0.8181 | 0.8135 | 0.8072 | 0.8207 | 0.8127 |
| kddcup09_appetency | binary | 0.9822 | 0.9822 | 0.9789 | 0.9648 | 0.9822 | 0.9817 |
| maternal_health_risk | multiclass | 0.8752 | 0.8374 | 0.8424 | 0.8358 | 0.8259 | 0.8374 |
| online_shoppers | binary | 0.9048 | 0.9029 | 0.8920 | 0.8869 | 0.9052 | 0.8975 |
| polish_bankruptcy | binary | 0.9673 | 0.9430 | 0.9337 | 0.9318 | 0.9411 | 0.9642 |
| qsar-biodeg | binary | 0.8973 | 0.8768 | 0.8705 | 0.8768 | 0.8720 | 0.8752 |
| seismic-bumps | binary | 0.9342 | 0.9342 | 0.9278 | 0.9291 | 0.9342 | 0.9291 |
| splice | multiclass | 0.9692 | 0.9645 | 0.9180 | 0.9190 | 0.9639 | 0.9655 |
| student_dropout | multiclass | 0.7827 | 0.7559 | 0.7341 | 0.7266 | 0.7563 | 0.7665 |
| taiwanese_bankruptcy | binary | 0.9729 | 0.9695 | 0.9668 | 0.9624 | 0.9697 | 0.9714 |
| website_phishing | multiclass | 0.8967 | 0.8918 | 0.8905 | 0.8782 | 0.8758 | 0.8745 |

## C. k-Sensitivity Evaluation

*Table 8.* k-sensitivity results reported as accuracy retention / predict-time speedup relative to Full TabICLv2.

| Dataset | $n_{\text{train}}$ | Full acc | Full pred (s) | $k = 16$ | $k = 32$ | $k = 64$ | $k = 128$ |
|---|---|---|---|---|---|---|---|
| credit-card-fraud | 227845 | 0.9996 | 96.47 | 99.99% / 4.59× | 100.00% / 3.47× | 100.00% / 2.24× | 100.00% / 1.29× |
| GiveMeSomeCredit | 120000 | 0.9373 | 26.94 | 99.81% / 3.33× | 99.87% / 2.28× | 99.86% / 1.25× | 99.84% / 0.68× |
| airline_satisfaction | 103904 | 0.9628 | 22.42 | 97.47% / 3.19× | 97.55% / 2.21× | 97.75% / 1.20× | 98.09% / 0.74× |
| jannis | 66986 | 0.7674 | 13.94 | 93.93% / 2.95× | 94.21% / 2.05× | 94.40% / 1.02× | 94.49% / 0.64× |
| APSFailure | 60800 | 0.9955 | 25.74 | 99.77% / 2.99× | 99.77% / 2.67× | 99.76% / 1.64× | 99.80% / 1.08× |
| adult | 39073 | 0.8762 | 4.18 | 97.88% / 1.97× | 98.36% / 1.26× | 98.66% / 0.67× | 98.90% / 0.29× |
| electricity | 30779 | 0.8844 | 2.59 | 96.96% / 1.66× | 96.92% / 1.03× | 96.88% / 0.60× | 97.15% / 0.25× |
| MagicTelescope | 10700 | 0.8864 | 0.65 | 96.15% / 1.23× | 96.29% / 0.77× | 96.63% / 0.44× | 97.47% / 0.23× |

## D. Batch-Inference Speed Evaluation

*Table 9.* Batch-inference speed evaluation.

| Dataset ($n_{\text{total}}$ / $n_{\text{train}}$) | Full (s) | Loc (s) | Speedup |
|---|---|---|---|
| credit-card-fraud (284.8k / 227.8k) | ∼99 | ∼28 | 3.50× |
| kddcup09_appetency (50.0k / 40.0k) | ∼63 | ∼24 | 2.67× |
| APSFailure (76.0k / 60.8k) | ∼55 | ∼21 | 2.59× |
| MiniBooNE (130.1k / 104.1k) | ∼51 | ∼20 | 2.52× |
| covertype (120.0k / 96.0k) | ∼50 | ∼20 | 2.50× |
| GiveMeSomeCredit (150.0k / 120.0k) | ∼44 | ∼19 | 2.29× |
| airline_satisfaction (129.9k / 103.9k) | ∼43 | ∼19 | 2.26× |
| jannis (83.7k / 67.0k) | ∼38 | ∼17 | 2.18× |
| Diabetes130US (71.5k / 57.2k) | ∼30 | ∼15 | 1.95× |
| numerai28.6 (96.3k / 77.1k) | ∼27 | ∼14 | 1.90× |
| SDSS17 (78.1k / 62.4k) | ∼20 | ∼13 | 1.57× |
| shuttle (58.0k / 46.4k) | ∼14 | ∼11 | 1.31× |
| adult (48.8k / 39.1k) | ∼13 | ∼10 | 1.26× |
| electricity (38.5k / 30.8k) | ∼11 | ∼11 | 1.03× |
| MagicTelescope (13.4k / 10.7k) | ∼8 | ∼10 | 0.77× |

# E. Main Small-Batch Latency Test

*Table 10.* Small-batch latency results for Full TabICLv2 and Localized TabICLv2. The table reports total prediction time, per-query latency, and speedup across different numbers of test queries.

| Dataset | $N_{train}$ | $N_{query}$ | Full total (s) | Loc total (s) | Full (ms/query) | Loc (ms/query) | Speedup |
|---|---|---|---|---|---|---|---|
| credit-card-fraud | 227,845 | 1 | 76.7540 | 0.0628 | 76754.02 | 62.767 | 1222.8× |
| credit-card-fraud | 227,845 | 5 | 76.7352 | 0.0633 | 15347.04 | 12.670 | 1211.3× |
| credit-card-fraud | 227,845 | 10 | 76.7377 | 0.0661 | 7673.77 | 6.608 | 1161.3× |
| credit-card-fraud | 227,845 | 50 | 76.7599 | 0.0778 | 1535.20 | 1.557 | 986.1× |
| credit-card-fraud | 227,845 | 100 | 76.8039 | 0.0903 | 768.04 | 0.903 | 850.1× |
| credit-card-fraud | 227,845 | 500 | 76.9406 | 0.2774 | 153.88 | 0.555 | 277.3× |
| GiveMeSomeCredit | 120,000 | 1 | 21.6351 | 0.0629 | 21635.15 | 62.859 | 344.2× |
| GiveMeSomeCredit | 120,000 | 5 | 21.6270 | 0.0624 | 4325.40 | 12.482 | 346.5× |
| GiveMeSomeCredit | 120,000 | 10 | 21.6301 | 0.0623 | 2163.01 | 6.227 | 347.4× |
| GiveMeSomeCredit | 120,000 | 50 | 21.6401 | 0.0726 | 432.80 | 1.452 | 298.1× |
| GiveMeSomeCredit | 120,000 | 100 | 21.7002 | 0.0830 | 217.00 | 0.830 | 261.4× |
| GiveMeSomeCredit | 120,000 | 500 | 21.7676 | 0.2336 | 43.54 | 0.467 | 93.2× |
| airline_satisfaction | 103,904 | 1 | 18.0945 | 0.0615 | 18094.47 | 61.477 | 294.3× |
| airline_satisfaction | 103,904 | 5 | 18.0934 | 0.0609 | 3618.68 | 12.171 | 297.3× |
| airline_satisfaction | 103,904 | 10 | 18.0923 | 0.0620 | 1809.23 | 6.198 | 291.9× |
| airline_satisfaction | 103,904 | 50 | 18.1090 | 0.0677 | 362.18 | 1.355 | 267.4× |
| airline_satisfaction | 103,904 | 100 | 18.0984 | 0.0795 | 180.98 | 0.795 | 227.6× |
| airline_satisfaction | 103,904 | 500 | 18.1685 | 0.2321 | 36.34 | 0.464 | 78.3× |
| numerai28.6 | 77,056 | 1 | 10.8834 | 0.0595 | 10883.38 | 59.537 | 182.8× |
| numerai28.6 | 77,056 | 5 | 10.8814 | 0.0593 | 2176.27 | 11.865 | 183.4× |
| numerai28.6 | 77,056 | 10 | 10.8821 | 0.0600 | 1088.21 | 6.004 | 181.3× |
| numerai28.6 | 77,056 | 50 | 10.8984 | 0.0637 | 217.97 | 1.275 | 171.0× |
| numerai28.6 | 77,056 | 100 | 10.8922 | 0.0750 | 108.92 | 0.750 | 145.3× |
| numerai28.6 | 77,056 | 500 | 10.9490 | 0.2187 | 21.90 | 0.437 | 50.1× |
| jannis | 66,986 | 1 | 11.3275 | 0.0557 | 11327.48 | 55.684 | 203.4× |
| jannis | 66,986 | 5 | 11.3455 | 0.0561 | 2269.10 | 11.227 | 202.1× |
| jannis | 66,986 | 10 | 11.3281 | 0.0554 | 1132.81 | 5.544 | 204.3× |
| jannis | 66,986 | 50 | 11.4461 | 0.0599 | 228.92 | 1.198 | 191.2× |
| jannis | 66,986 | 100 | 11.4492 | 0.0734 | 114.49 | 0.734 | 156.0× |
| jannis | 66,986 | 500 | 11.4688 | 0.2271 | 22.94 | 0.454 | 50.5× |
| adult | 39,073 | 1 | 3.3822 | 0.0549 | 3382.18 | 54.886 | 61.6× |
| adult | 39,073 | 5 | 3.3827 | 0.0563 | 676.54 | 11.258 | 60.1× |
| adult | 39,073 | 10 | 3.3862 | 0.0564 | 338.62 | 5.640 | 60.0× |
| adult | 39,073 | 50 | 3.3893 | 0.0608 | 67.79 | 1.215 | 55.8× |
| adult | 39,073 | 100 | 3.3896 | 0.0700 | 33.90 | 0.700 | 48.4× |
| adult | 39,073 | 500 | 3.4239 | 0.1977 | 6.85 | 0.395 | 17.3× |

# F. Controlled Scaling Test: credit-card-fraud

*Table 11.* Controlled latency scaling results on the credit-card-fraud dataset. The table varies the number of training rows and test queries, and reports total prediction time, per-query latency, and speedup for Full TabICLv2 versus Localized TabICLv2.

| $N_{\text{train}}$ | $N_{\text{test}}$ | Full total (s) | Loc total (s) | Full (ms/query) | Loc (ms/query) | Speedup |
|---|---|---|---|---|---|---|
| 10,000 | 1 | 0.7432 | 0.0524 | 743.16 | 52.377 | 14.2× |
| 10,000 | 10 | 0.7434 | 0.0524 | 74.34 | 5.245 | 14.2× |
| 10,000 | 100 | 0.7503 | 0.0629 | 7.50 | 0.629 | 11.9× |
| 25,000 | 1 | 2.2498 | 0.0538 | 2249.81 | 53.849 | 41.8× |
| 25,000 | 10 | 2.2510 | 0.0548 | 225.10 | 5.479 | 41.1× |
| 25,000 | 100 | 2.2660 | 0.0664 | 22.66 | 0.664 | 34.1× |
| 50,000 | 1 | 5.9887 | 0.0560 | 5988.67 | 56.046 | 106.9× |
| 50,000 | 10 | 5.9894 | 0.0580 | 598.94 | 5.805 | 103.2× |
| 50,000 | 100 | 6.0252 | 0.0704 | 60.25 | 0.704 | 85.5× |
| 100,000 | 1 | 18.0431 | 0.0585 | 18043.08 | 58.450 | 308.7× |
| 100,000 | 10 | 18.0225 | 0.0592 | 1802.25 | 5.924 | 304.2× |
| 100,000 | 100 | 18.0554 | 0.0782 | 180.55 | 0.782 | 230.8× |
| 200,000 | 1 | 60.5042 | 0.0620 | 60504.22 | 62.039 | 975.3× |
| 200,000 | 10 | 60.5045 | 0.0644 | 6050.45 | 6.440 | 939.4× |
| 200,000 | 100 | 60.5301 | 0.0869 | 605.30 | 0.869 | 696.9× |
| 227,845 | 1 | 76.7734 | 0.0632 | 76773.37 | 63.244 | 1213.9× |
| 227,845 | 10 | 76.7950 | 0.0661 | 7679.50 | 6.605 | 1162.6× |
| 227,845 | 100 | 76.8629 | 0.0906 | 768.63 | 0.906 | 848.6× |

# G. ICL Head Ablation

*Table 12.* ICL head ablation results comparing Localized TabICL with Stage 2+3 fine-tuning against retrieval-only baselines. Binary tasks report ROC AUC, where higher is better, and multiclass tasks report log-loss, where lower is better. XGB-kNN denotes XGBoost trained on the retrieved k-nearest-neighbour context, and kNN-MV denotes k-nearest-neighbour majority vote.

| Dataset | Task | Loc-FT | XGB-kNN | kNN-MV | Loc>XGB-kNN | Loc>kNN-MV |
|---|---|---|---|---|---|---|
| APSFailure | binary | 0.9911 | 0.9779 | 0.9704 | ✓ | ✓ |
| Amazon_employee_access | binary | 0.8247 | 0.6262 | 0.5876 | ✓ | ✓ |
| Bank_Customer_Churn | binary | 0.8409 | 0.6735 | 0.5708 | ✓ | ✓ |
| Bioresponse | binary | 0.8082 | 0.8258 | 0.7775 | ✗ | ✓ |
| Diabetes130US | binary | 0.6390 | 0.4828 | 0.4653 | ✓ | ✓ |
| E-CommerceShipping | binary | 0.7392 | 0.7356 | 0.7028 | ✓ | ✓ |
| Fitness_Club | binary | 0.8079 | 0.7748 | 0.7806 | ✓ | ✓ |
| GiveMeSomeCredit | binary | 0.8460 | 0.7673 | 0.7676 | ✓ | ✓ |
| HR_Analytics | binary | 0.7864 | 0.7254 | 0.7392 | ✓ | ✓ |
| Is_good_customer | binary | 0.7305 | 0.6562 | 0.6470 | ✓ | ✓ |
| MIC | multiclass | 0.5943 | 5.2410 | 2.0717 | ✓ | ✓ |
| Marketing_Campaign | binary | 0.8881 | 0.7591 | 0.7480 | ✓ | ✓ |
| NATICUSdroid | binary | 0.9860 | 0.9823 | 0.9802 | ✓ | ✓ |
| SDSS17 | multiclass | 0.1043 | 0.8802 | 0.8770 | ✓ | ✓ |
| airline_satisfaction | binary | 0.9874 | 0.9320 | 0.8439 | ✓ | ✓ |
| anneal | multiclass | 0.0243 | 7.1132 | 0.7424 | ✓ | ✓ |
| bank-marketing | binary | 0.7412 | 0.6898 | 0.7027 | ✓ | ✓ |
| blood-transfusion | binary | 0.7244 | 0.7234 | 0.7603 | ✓ | ✗ |
| churn | binary | 0.9150 | 0.7165 | 0.7065 | ✓ | ✓ |
| coil2000 | binary | 0.7034 | 0.6555 | 0.6655 | ✓ | ✓ |
| credit-g | binary | 0.7437 | 0.6594 | 0.6230 | ✓ | ✓ |
| credit_card_default | binary | 0.7737 | 0.6577 | 0.6513 | ✓ | ✓ |
| diabetes | binary | 0.8320 | 0.7787 | 0.7107 | ✓ | ✓ |
| hazelnut_contaminant | binary | 0.9731 | 0.9394 | 0.8770 | ✓ | ✓ |
| heloc | binary | 0.7670 | 0.7075 | 0.7346 | ✓ | ✓ |
| hiva_agnostic | multiclass | 0.2097 | 1.0253 | 0.8260 | ✓ | ✓ |
| in_vehicle_coupon | binary | 0.8282 | 0.7710 | 0.7076 | ✓ | ✓ |
| jm1 | binary | 0.7627 | 0.6828 | 0.6762 | ✓ | ✓ |
| kddcup09_appetency | binary | 0.7409 | 0.5812 | 0.5850 | ✓ | ✓ |
| maternal_health_risk | multiclass | 0.3949 | 3.2352 | 0.8957 | ✓ | ✓ |
| online_shoppers | binary | 0.9283 | 0.8565 | 0.8655 | ✓ | ✓ |
| polish_bankruptcy | binary | 0.8800 | 0.7071 | 0.7142 | ✓ | ✓ |
| qsar-biodeg | binary | 0.9345 | 0.9069 | 0.8582 | ✓ | ✓ |
| seismic-bumps | binary | 0.7481 | 0.7190 | 0.7331 | ✓ | ✓ |
| splice | multiclass | 0.1241 | 0.7585 | 0.7440 | ✓ | ✓ |
| student_dropout | multiclass | 0.6119 | 1.0759 | 0.9252 | ✓ | ✓ |
| taiwanese_bankruptcy | binary | 0.9409 | 0.6429 | 0.6346 | ✓ | ✓ |
| website_phishing | multiclass | 0.2783 | 0.9305 | 0.4647 | ✓ | ✓ |
| Win rate | | | | | 37/38 | 37/38 |

