# OpenReview forum: "Localized TabICLv2: Scaling Tabular In-Context Learning through k-NN"
_ICML.cc/2026/Workshop/FMSD — FMSD @ ICML 2026 Poster_

### Official Review · Reviewer_T2hJ · 2026-05-20
**Review of Localized TabICLv2: Scaling Tabular In-Context Learning through k-NN**

**Rating:** 6
**Confidence:** 4

**Review:**

**Recommendation: Weak accept.** The research question is relevant, but the contribution and results are limited.

**Summary**

The paper proposes Localized TabICLv2, a method to reduce the inference cost of TabICLv2. Instead of letting each test point attend to the full training set during in-context learning, it retrieves only the most similar training rows per test point and runs ICL on that smaller local context. The main novelty is that the retrieval is performed in the model's learned row representation space rather than in the raw input feature space, which the authors show produces better neighbours for the downstream ICL task. They additionally fine-tune the relevant stages of the model to better adapt to this localized setting. They evaluate on TabArena and report that the localized model retains most of the accuracy of the full model while achieving a substantial speedup, especially in single-query serving.

**Strengths**

- The research question is relevant: inference cost is a real bottleneck for tabular foundation models, and any work that makes them more practical to deploy is useful to the community.
- The evaluation follows the standard TabArena protocol, which makes the numbers easy to interpret and compare against other work.

**Weaknesses**

- The method is essentially a port of an existing LLM technique (k-NN retrieval to limit attention context) to TabICLv2. The contribution is mostly in the integration rather than in a new mechanism.
- The scope is narrow: the method is tied to the specific three-stage architecture of TabICLv2, which limits broader applicability.
- The reported improvement from changing the retrieval space (raw features → learned row embeddings) is marginal, which weakens the main contribution.

**Suggestions**

- A comparison against a random-k baseline (e.g. k=32 randomly drawn training rows) is missing. Without this, it is hard to tell whether the accuracy retention comes from the k-NN retrieval being meaningfully informative or simply from the ICL head being robust to small context sizes. This baseline is cheap to add and would directly support the central claim of the paper.
- For a fair latency comparison, you might also want to compare against persisting KV-caches to disk and loading them on demand. The current text of "Limited-Query Latency" suggests the full attention is computed for every prediction call instead of the KV-cache being loaded. KV-cache disk offloading is now standard in production LLM serving (e.g. vLLM with LMCache, llm-d), where caches survive restarts and are reused across queries. This would be a more realistic deployment baseline for the single-query serving setting.
- If you are already porting LLM techniques to speed up inference, distilling TabICLv2 into a smaller non-ICL variant is a natural secondary direction worth exploring. It would be informative to know whether retrieval-based localization gives more than distillation for the same accuracy budget.

---

### Official Review · Reviewer_dN98 · 2026-05-22
**Clean acceleration recipe for TabICLv2 with strong single-query latency evidence, weakened by a missing LoCalPFN-on-TabICLv2 baseline and under-specified fine-tuning protocol.**

**Rating:** 6
**Confidence:** 4

**Review:**

## *Summary*

The paper restricts TabICLv2's Stage 3 in-context predictor to the k cosine-nearest training rows in TabICLv2's own Stage 2 representation space, then fine-tunes Stages 2 and 3 jointly (Stage 1 frozen) on mix_scm synthetic tables with sequence lengths 10k–30k and k fixed at 32. On 38 TabArena classification tasks, the fine-tuned localized model retains most of the full-context accuracy while improving inference speed substantially, especially in the single-query regime. A k-sensitivity sweep shows k=32 balances accuracy retention (97.57% before FT) and speedup (1.97× mean). Retrieval in the Stage 2 embedding space beats raw-feature retrieval (Embed-FT vs Raw-FT). An ICL-head ablation against XGBoost-on-k-NN and k-NN majority vote, both using the same retrieved k=32, shows the localized model winning on 37/38 datasets, indicating the ICL predictor contributes beyond raw neighbour aggregation.

## *Strengths*

- Deployment-relevant framing. The single-query latency result (~59 ms vs ~76 s on credit-card-fraud at N≈228K) is the number practitioners care about for online scoring, and the controlled scaling study on credit-card-fraud (Appendix F) is the kind of evidence that the space needs.
- The k-sensitivity sweep (k ∈ {16, 32, 64, 128}, Table 1) gives an interpretable operating-point trade-off. k=128 reaches 98.90% retention at 0.66× (slower than full context), k=16 reaches 96.93% retention at 2.77×. The k=32 default is justified on the sweep.
- The ICL-head ablation (Section 4, Table 5) tests the obvious counter-hypothesis: maybe k-NN alone explains the accuracy. Winning 37/38 against XGBoost-on-k-NN and k-NN majority vote on the same retrieved context strongly suggests that the ICL head contributes beyond simple neighbour aggregation.
- The Embed-vs-Raw retrieval comparison (Table 2) defends the architectural choice to retrieve in Stage 2 rather than raw features. Embed-FT beats Raw-FT on AUC (0.827 vs 0.817), log-loss (0.293 vs 0.319), and accuracy (0.873 vs 0.872).
- Per-dataset appendix tables are unusually transparent for a 4-page workshop submission. Tables 6–12 allow independent verification.

## *Areas for Improvement*

- The closest prior work, LoCalPFN (Thomas et al., NeurIPS 2024), is acknowledged but not compared empirically. LoCalPFN performs k-NN retrieval on TabPFN; the paper argues its retrieval is not optimized for the downstream ICL task. That methodological difference is real, but a head-to-head with LoCalPFN-style retrieval applied to TabICLv2 (no S2+S3 fine-tuning) on the same 38 datasets is needed to attribute the 1.18 pp recovery to fine-tuning rather than to TabICLv2's stronger base representations.
- The fine-tuning procedure is under-specified. It is unclear whether k-NN retrieval is performed at every training step (matching inference) or whether the model trains on full context and is retrieval-restricted only at inference. The former is the natural reading but should be stated explicitly, since it determines whether deploying with k ≠ 32 requires retraining.
- Some of the retention numbers may be harder to interpret on datasets where both the full and localized models already appear close to a ceiling. On at least four datasets (anneal, hiva_agnostic, kddcup09_appetency, seismic-bumps), full TabICL-v2 and the FT-localized model report identical accuracy in Table 7, suggesting both hit a class-imbalance ceiling. A retention breakdown on datasets where the full model meaningfully exceeds the majority-class baseline would sharpen the claim.
- Retrieval cost is folded into the localized end-to-end time without isolation. Appendix A acknowledges T_retrieval but the speedup tables do not separate it. For very large training sets, cosine similarity over N_train rows becomes non-trivial.
- k is hard-coded to 32 throughout fine-tuning. No experiment varies k during FT. If a deployed system needs k=64 for higher accuracy, it is unclear whether the FT model generalizes or whether re-fine-tuning is required.
- Classification only. TabICLv2 supports both classification and regression. A small regression sanity check would strengthen generality.

## *Detailed Comments*

1. The single-query speedup table (Table 4) reports "approximate" speedups; per-dataset values in Appendix E vary by an order of magnitude (61× to 1223× at N_test=1). The 249× median is the right summary, but the abstract phrasing reads as a typical case rather than the median of a wide distribution.
2. The k-sensitivity sweep uses 8 datasets; the main TabArena evaluation uses 38. Different subsets, so the k=32 choice is justified on the smaller set and deployed on the larger.
3. The "fine-tuned model outperforms the pretrained localized variant on 33/38" win-count statement is not paired with a sign test or paired Wilcoxon. With mean improvement ≈ 0.012 accuracy, adding one is straightforward.
4. On Bioresponse (the one ICL-ablation loss), the localized model trails XGBoost-on-k-NN by 0.018 AUC. A brief explanation (high feature count? noisy local neighbourhoods?) would round out the 37/38 framing.
5. Appendix A's theoretical formula T_S3_full / T_S3_local ≈ N_train(N_train + N_test) / (N_test · k²) is correct as written but rests on full quadratic attention. FlashAttention-style implementations would change the constant.

## *Justification of Score*

The paper executes a known acceleration idea cleanly on a current-SOTA tabular model, and the empirical work is honest. The k-sensitivity sweep, ICL-head ablation, and Embed-vs-Raw comparison are the right experiments, and the single-query latency result is directly relevant to the workshop's scope.

Two issues hold the score back. It is still somewhat unclear to me on how much of the gain comes from the localization+fine-tuning recipe itself versus the strength of the underlying TabICLv2 representations. Without a LoCalPFN-on-TabICLv2 baseline, the central methodological claim that joint S2+S3 fine-tuning matters rests on one ablation rather than a controlled comparison. The fine-tuning recipe under-specification is fixable in revision but currently leaves a reproducibility gap.

---

### Official Review · Reviewer_MF2L · 2026-05-22

**Rating:** 5
**Confidence:** 3

**Review:**

The paper proposes Localized TabICLv2, a faster version of TabICLv2 for large tabular datasets. It retrieves the top-k nearest training rows using Stage 2 row embeddings, then runs Stage 3 ICL only on those local neighbors instead of the full training set. The main contribution is practical: with k = 32 and Stage 2+3 fine-tuning, it keeps most of Full TabICLv2 accuracy while giving almost 2× median batch speedup .

Strengths : The paper targets a real problem: full-context TabICLv2 inference becomes expensive as the training set grows. The method is simple and easy to apply, since it does not require major architectural changes. The ablations are useful: embedding-space retrieval beats raw-feature retrieval, and the ICL head beats retrieval-only baselines on 37/38 datasets.

Areas for Improvement : The novelty is limited. k-NN localized tabular ICL has already been explored in prior work such as LoCalPFN ; this paper mainly adapts the idea to TabICLv2. The paper mentions LoCalPFN but does not compare against it experimentally. This is the biggest missing baseline. The speedup claim needs careful framing. The 249× result is mainly for single-query serving; the more general batch speedup is much smaller at about 2.18× median. The evaluation only covers classification on 38 TabArena datasets. Regression is missing, even though TabICLv2 supports regression.

Detailed Comments : Add a direct LoCalPFN comparison, or explain clearly why it is infeasible. Explore adaptive k instead of using fixed k = 32 for all datasets and queries. Report GPU memory usage, not just runtime, since memory scaling is part of the motivation. Add failure analysis for datasets where localization loses more accuracy. Discuss approximate k-NN / FAISS-style retrieval for very large datasets. Report the compute cost of Stage 2+3 fine-tuning.

NOTE : The main new element is Stage 2 embedding-based k-NN plus Stage 2+3 fine-tuning for localized TabICLv2 inference. The localization idea itself is incremental, but the adaptation to TabICLv2 is practical and technically sound.

Justification of Score : The work is technically sound and practically useful, especially for low-latency single-query serving. However, the contribution is incremental. The main idea already exists in retrieval-localized tabular ICL, and the missing LoCalPFN comparison weakens the novelty claim.

https://github.com/layer6ai-labs/LoCalPFN